# Lactic Starter Dose Shapes *S. aureus* and STEC O26:H11 Growth, and Bacterial Community Patterns in Raw Milk Uncooked Pressed Cheeses

**DOI:** 10.3390/microorganisms9051081

**Published:** 2021-05-18

**Authors:** Justine Piqueras, Christophe Chassard, Cécile Callon, Etienne Rifa, Sébastien Theil, Annick Lebecque, Céline Delbès

**Affiliations:** 1UMR 0545 Fromage, Université Clermont Auvergne, INRAE, VetAgro Sup, 20 Côte de Reyne, F-15000 Aurillac, France; justine.piqueras@inrae.fr (J.P.); christophe.chassard@inrae.fr (C.C.); cecile.callon@inrae.fr (C.C.); sebastien.theil@inrae.fr (S.T.); annick.lebecque@orange.fr (A.L.); 2UMR INSA/INRA 792, Toulouse Biotechnology Institute, INSA/CNRS 5504, 135 Avenue de Rangueil, F-31077 Toulouse, France; etienne.rifa@insa-toulouse.fr

**Keywords:** cheese, raw milk, lactic starter, STEC, CoPS, bacterial diversity, metagenomics

## Abstract

Adding massive amounts of lactic starters to raw milk to manage the sanitary risk in the cheese-making process could be detrimental to microbial diversity. Adjusting the amount of the lactic starter used could be a key to manage these adverse impacts. In uncooked pressed cheeses, we investigated the impacts of varying the doses of a lactic starter (the recommended one, 1×, a 0.1× lower and a 2× higher) on acidification, growth of *Staphylococcus aureus* SA15 and Shiga-toxin-producing *Escherichia coli* (STEC) O26:H11 F43368, as well as on the bacterial community patterns. We observed a delayed acidification and an increase in the levels of pathogens with the 0.1× dose. This dose was associated with increased richness and evenness of cheese bacterial community and higher relative abundance of potential opportunistic bacteria or desirable species involved in cheese production. No effect of the increased lactic starter dose was observed. Given that sanitary criteria were paramount to our study, the increase in the pathogen levels observed at the 0.1× dose justified proscribing such a reduction in the tested cheese-making process. Despite this, the effects of adjusting the lactic starter dose on the balance of microbial populations of potential interest for cheese production deserve an in-depth evaluation.

## 1. Introduction

An increasing part of consumers are looking for traditional products, with distinctive sensorial characteristics [1,2], such as raw milk cheeses. The sensorial characteristics of raw milk cheeses are strongly related to the richness of microbiota [3,4,5,6,7], itself dependent on the various initial sources of enrichment [1,5,8,9].

Apart from the variation of sensorial characteristics [1], using raw milk for cheese manufacture can also lead to an increased risk of contamination by food spoilage or pathogenic bacteria [10]. Unwanted bacteria can be responsible for the development of unpleasant flavors or visual defects like slits [11], causing an economic loss for the dairy industry. They can also be involved in sanitary risks. Because of their role in Collective Food Toxi-Infection (CFTI), major pathogens, such as *Salmonella* spp., *Listeria monocytogenes*, Shiga-toxin-producing *Escherichia coli* (STEC) and Coagulase Positive *Staphylococcus* (CoPS), are targeted by controls in food including raw milk cheeses [12].

Worldwide, *Staphylococcus aureus* is the most prevalent pathogen responsible for foodborne intoxication. In France, in 2018, it was the most frequently detected or suspected agent responsible for CFTI and represented 3% of CFTI related to the consumption of dairy products [13], with STEC representing 2.5% of the sources of CFTI [13]. STEC can cause severe hemolytic uremic syndrome, with O26:H11 being the most commonly found serotype in dairy products, including raw milk cheese [14]. Because of the high prevalence of *S. aureus* in dairy products, and the severity of the symptoms associated with STEC infections, these two pathogens are subjected to high vigilance.

Sanitary risk levels also vary according to the cheese type. Thus, due to their short ripening duration, high activity of water and also their slow or moderate acidification, uncooked pressed cheeses present an increased risk for the two previously mentioned pathogens. In these types of cheese, managing acidification by reaching targeted values of pH is a key to controlling sanitary risk [12,15,16] in that it allows for by allowing for an improvement in the protection against the growth of *S. aureus* and STEC.

Acidification is primarily driven by lactic acid bacteria (LAB) that produce lactic acid by lactose degradation. Thus, to ensure early acidification in the cheese-making processes, the milk can be inoculated with starter cultures composed of LAB [10].

Besides traditional inoculation practices, like back-slopping [5,17], multiple commercial solutions have been developed to offer starters adapted to each cheese type [7,10]. These lactic starters are composed of LAB that are generally recognized as safe (GRAS). They are selected for their cheese origin, their synergistic activities with other LAB, as well as their abilities to produce metabolites of interest, from antimicrobial compounds to the metabolites involved in sensorial characteristics. The addition of lactic starters allows for a more constant quality of cheese products [1,10,18], thus reducing the loss of products due to cheese defects. The use of commercial lactic starters has therefore become more and more widespread.

Microbial culture is relevant to evaluate the levels of the main cultivable microbial groups in cheeses and remains the basis of the standard methods for the determination of pathogens in foods (NF EN ISO 6888-2 [19], and ISO/TS 13136:2012 [20]). For more than twenty years, the development of culture-independent methods, based on the sequencing of universally conserved and hypervariable regions of 16S ribosomal DNA, has brought new knowledge about cheese microbial ecosystems, allowing for the rapid identification of the whole bacterial DNA from cheese environments [21]. Thus, such methods allow us to compare the microbial communities between different cheeses [22] or specify their spatial or temporal distribution [23]. Overall, these methods allow us to better understand the importance of the microbial diversity and dynamics in cheeses and to identify their main environmental sources and technological drivers [5]. Potentially negative impacts of the use of lactic starters on the microbial diversity in cheese products have been revealed. Indeed, the use of a lactic starter inoculated at high levels impacts the levels of autochthonous species [24,25]. During the cheese-making process, LAB species from a lactic starter become dominant [26]. When the starter used was produced by a natural method like back-slopping, the dominance of the LAB species identified in the natural lactic starter was also observed [27]. Reduced bacterial diversity, as well as lower richness and evenness, were observed in cheeses manufactured with a lactic starter, compared with cheeses manufactured without the addition of a lactic starter [28].

To our knowledge, only a few studies have looked into the impact of changing the inoculation level of the lactic starter. As an example, Fernandez-García et al. [29] and Khosrowshahi et al. [30] observed that, in Hispanico cheeses (semi-hard) and Iranian White cheeses (close-textured brined cheeses), respectively, the higher the dose of a mesophilic starter, the lower the pH values reached within the first 24 h of cheese-making. On the contrary, Özer et Kesenkaş, [31], did not observe any differences in the pH values in conjunction with the lactic starter dose on the first day in Mihalič cheese (a Turkish hard cheese stored in brine). Among these authors, only Fernandez-García et al., [29] looked at the impacts on the microbial characteristics from the variations in the dose of the lactic starter culture, but they did not observe any modification with regards to the lactic starter dose on mesophilic, Gram-negative bacteria or on lactobacilli counts.

Consequently, adjusting the lactic starter to the necessary and sufficient dose could be a key to improving cheese production, to managing sanitary risks while promoting microbial diversity.

In this regard, the principal aim of our study was to investigate first whether a reduced dose of lactic starter, compared with the recommended dose, would satisfy sanitary criteria while minimizing the effects on the microbial diversity. We also investigated whether the protection against pathogens could be improved using a higher lactic starter dose, and at what cost for the bacterial diversity. Taking the dose recommended by the lactic starter manufacturer as a reference, we also simultaneously tested a 10-fold lower (0.1×) and a 2-fold higher (2×) dose. These were added into raw milk originating from two different farms. The effects of these three different doses on the growth of *S. aureus* SA15 and STEC O26:H11 F43368, as well as on the microbial community composition, were assessed in uncooked pressed cheese models.

## 2. Materials and Methods

### 2.1. Strains and Culture Conditions

A unique batch of the lyophilized commercial culture MY800 (*Streptococcus thermophilus*, *Lactobacillus delbrueckii* spp. *bulgaricus*, *Lb. delbrueckii* subsp. *lactis*, Danisco, Copenhagen, Denmark) was used as a lactic starter culture for all cheese manufactures. The lyophilized lactic starter was stored at−20 °C before use. The recommendation for use is 5 Danisco Culture Units (DCU) for each 100 L of milk. The lyophilized lactic starter was revived in sterile reconstituted skimmed milk (LACTALIS Ingredients, Bourgbarré, France) for 2 h 30 min at 42 °C, and then was stored at 4 °C until use within 3 days.

STEC O26:H11 strain F43368 and coagulase (+) *S. aureus* strain SA15, isolated from “tome de pays” in 2009, and raw milk, respectively, were provided by UMRF (Unité Mixte de Recherche sur le Fromage, INRAE, Aurillac, France). Both pathogenic strains were revived in Brain Heart Infusion (BHI) broth (BIOKAR Diagnostics, Pantin, France), for 24 h at 37 °C. The strain of STEC O26:H11 was enumerated on BHI agar medium. The *S. aureus* strain was enumerated on Baird-Parker medium, supplemented with Rabbit Plasma Fibrinogen (RPF medium) (BIOKAR Diagnostics, Pantin, France) [32]. Aliquots of known concentrations were stored at −80 °C, until use in the cheese-making assays, 2 weeks after storage for the last. Inoculated cheeses samples on D1 were tested for the presence of staphylococcal enterotoxins by the Enzyme Linked Fluorescent Assay based Vidas Set 2 system (AgroLab’s, Aurillac, France), which targets staphylococcal enterotoxins A to E with a limit of detection of 0.25 ng toxin per gram of food, according to NF EN ISO 19020:2017 [33].

### 2.2. Experimental Design

In January 2019, raw milk was obtained from two different farms (“F_15” and “F_38”), located no more than 20 km from the cheese-making plant. It was collected early in the morning from a refrigerated milk tank on the farm and used within 1 h to produce Saint-Nectaire-like uncooked pressed cheeses of reduced size (500 g) at the UMRF experimental cheese plant.

On each day of the cheese-making process, the milk from both farms was tested under the same conditions of inoculation, and each pair was repeated three times, over a 2-week period. Two cheeses were made from each batch of milk: a “control” cheese, without pathogen inoculation, and an “assay” cheese, with controlled inoculation of pathogens.

Pathogens were inoculated into the milk prior to the cheese-making process. The strains *S. aureus* SA15 and STEC O26:H11 F43368 were inoculated at the expected levels of 10^3^ CFU/mL and 10^2^ CFU/mL, respectively, together in the assay milk vats. In control vats, no pathogen was added.

Both the assay and the control vats were inoculated either with the recommended dose of lactic starter MY800 (6 × 10^6^ CFU/g, referred to as “1×”), with a 10-fold lower dose (6 × 10^5^ CFU/g, referred to as “0.1×”) or with a 2-fold higher dose (1 × 10^7^ CFU/g, referred to as “2×”).

### 2.3. Cheese-Making and Sampling

Vats were filled with 5 L of raw milk, and all the vats were heated at 33 °C. The lactic starter (MY800) was inoculated at the chosen dose in each vat and then these vats were inoculated with pathogens. After being mixed for 15 min, the milk was inoculated with the ripening starter (30 mL per 500 L of milk, composed of *Brevibacterium linens*, *Debaryomyces hansenii*, *Penicillium fuscoglaucum*, and *Fusarium domesticum* (SN MEL N27 500 L, LIP Aurillac)), and rennet (40 mL per 100 L of milk, containing 520 mg/L of active chymosin (Fabre LCP Food Ingredients, Prat, France)). The inoculated milk was processed according to the uncooked pressed cheese technology as previously described [34,35] and salted at 5% with coarse salt. The cheeses were then ripened in UMRF’s ripening cellars at 8 °C and 95% relative humidity for 28 days. During the ripening, the cheeses were washed at 6 and 9 days of ripening with the solution of the ripening starter. They were turned over once a week.

The raw milk was sampled before the cheese was made to evaluate the bacterial and chemical composition and again after inoculation to control the dose of the lactic starter and the level of pathogens. Cheese cores were sampled after 1 (D1), and 8 (D8) days. After 28 days (D28), a last sampling was realized during which the rind (with a thickness of 1.5 mm) and the core from the same cheese were separately sampled. At each sampling, a 100 g-piece of cheese was taken and separated into equivalent portions intended for microbiological, metagenomic and biochemical analyses. The sample dedicated to the metagenomic analyses was roughly sliced with a knife before storage at −20 °C.

### 2.4. Physico-Chemical Analyses

Several physico-chemical parameters were determined in the fresh raw milk (AgroLab’s, Aurillac, France). Urea, fat, protein, lactose and casein levels, as well as fatty acid profiles, were estimated by infra-red-light spectrometry. Somatic cell counts were estimated by epifluorescence and the lipolysis rate by Copper Soap method.

pH measures were carried out with a 926 VTV pH-meter with Ingold electrode 406 MX (Mettler-Toledo S.A., Viroflay, France), in the milk, on the fresh cheese at 6 h (D0.25) and at the time of sampling (D1 and D28). They were taken from three different points on each sampled cheese, at room temperature.

The dry matter content was calculated in the milk, in one-day-old cheese (D1, at the entry in the cellar) and at the end of ripening, according to ISO 5534:2004 [IDF 4:2004] [36].

The rheological method used was uniaxial compression at a constant displacement rate, as described by Frétin et al. [37]. Resistance to penetration measures of the cheese core piece were taken on seven points uniformly distributed over the surface of the control cheese sample. The Young’s modulus (stress/relative strain), proportional to cheese firmness and inversely related to cheese elasticity and creaminess [38], was calculated.

### 2.5. Microbiological Analyses

Ten grams of cheese were ground in phosphate buffer solution using a homogenizer mill (BagMixer, Interscience, Saint Nom la Brétèche, France), as described by Frétin et al. [37]. To evaluate the levels of microbial groups, both the raw and inoculated milk, as well as the ground cheese, were plated directly without or with dilution in Ringer’s solution. Inoculations were carried out with EasySpiral (Interscience) in exponential mode, with serial dilutions in Ringer’s solution.

In the fresh raw milk, the levels of *E. coli* and total microorganisms were evaluated by AgroLab’s, by counting on TBX selective media and by flow cytometry (Bactoscan FC, FOSS, Hilleroed, Denmark), respectively, while the level of autochthonous coagulase positive *Staphylococcus* was estimated by counting on an RPF medium, and the absence of autochthonous STEC was verified by enrichment in peptone water followed by plating on ChromID coli at 42 °C (Biomerieux, Marcy l’Etoile, France).

After revival in sterile reconstituted skimmed milk, the lactic starter was enumerated on M17 agar [39] at 42 °C, for 24 h to determine the volume needed to inoculate the milk at the desired dose during the manufacturing of the cheese. Both batches of inoculated milk were then verified for the levels of *Streptococcus*, STEC O26:H11 and *S. aureus*, on M17 agar at 42 °C, ChromID coli at 42 °C, and RPF medium at 37 °C, respectively. The levels of the pathogens were controlled in the cheese cores, at D1 and D28, by plating on the same media. As well, the level of thermophilic streptococci was also checked at D28 in the cheese cores and in the rinds. The levels of the other microbial groups such as lactic and ripening bacteria, yeasts and molds were evaluated by counting on MRS agar [40] at 30 °C, CRBM agar [41] at 25 °C, and OGA medium [42] at 25 °C, respectively, at D28 in the core and the rind of the cheeses.

### 2.6. Amplicon Sequencing

The milk (100 mL) was centrifuged at 5285× *g* for 30 min, at 4 °C (Sigma 6K15, Kanagawa, Japan). The fat and the supernatant were discarded, and the pellet was resuspended with 1mL of PBS. After a 5-min centrifugation at 13,000× *g* (Eppendorf 5415D, Hamburg, Germany), the supernatant was removed, and the pellet was stored at −20 °C until extraction. A thawed, two-gram sample of the control cheese was ground with a homogenizer mill (BagMixer, Interscience, Saint Nom la Brétèche, France) with 2 mL of 4 M guanidine thiocyanate −0.1 M Tris (pH 7.5) and 500 μL of the mixture were transferred into a 2 mL tube, on which an extraction was performed.

Extractions were performed with a series of 24 sample tubes including one blank extraction tube, in which ground cheese or the milk pellet were replaced by PCR grade water (Merck KGaA, Darmstadt, Germany). 40 µL of 10% N-lauroylsarcosine and 200 mg of zirconium beads were added to each tube and then mixed with a bead mill homogenizer (Precellys Evolution, Bertin Technologies SAS, Ozyme, France) (one 20-s run at a speed of 6500 m/s). The following steps were the same as described by Frétin et al. [34], except the final precipitation which was performed as described by Duthoit et al. [43].

The DNA was quantified using the Qubit Fluorometric Quantitation (Invitrogen, Thermo Fisher Scientific, Waltham, MA, USA) method with a High Sensitivity kit for DNA extracts from milk and a Broad Range kit for DNA extracts from cheeses (rind and core).

The variable region V3–V4 of the 16S rRNA gene (~510 bp) was targeted using the primers MSQ-16SV3F (5′-TACGGRAGGCWGCAG-3′) [44] and PCR1R-460 (5′-TTACCAGGGTATCTAATCCT-3′) and amplified as previously described by Frétin et al. [8]. The amplified products were used for Illumina paired-end library preparation and cluster generation, followed by 250 bp paired-end sequencing on an Illumina MiSeq instrument (INRAE, GeT-PLaGE platform, Toulouse, France).

The raw data, provided by sequencing, were processed with the pipeline rANOMALY [45], which relies on the dada2 R package [46] to produce amplicon sequence variants (ASV) as taxonomic units. A step of decontamination was carried out, based both on the prevalence of contaminant ASVs, as identified in the blank samples, and on the DNA concentration, as described by Theil et Rifa [45]. The filtered ASV count table was used to perform statistical analyses.

### 2.7. Statistical Analyses

Significance was declared at *p* < 0.05. *p*-values between 0.05 and 0.10 were considered a tendency, while *p* > 0.10 was considered not significant. Statistical analyses were performed with R for Windows (version 3.6.3, R Foundation for Statistical Computing, Vienna, Austria). The packages “dunn.test”, “ggplot2” were used for pairwise comparison of the lactic starter dose. Pathogen counts and pH values were analyzed with a Dunn’s test, allowing multiple comparisons using rank sum, with a Bonferroni’s correction. The farm effect was tested by a Wilcoxon’s test. The absence of interaction with the farm was confirmed with an ANOVA, where the variables, lactic starter dose, farm and their interactions, were the fixed factors.

To study how modulating the dose of the lactic starter impacts the bacterial communities in raw milk-cheeses, the rANOMALY interface was used. To compare evenness and richness (α-diversity) between tested conditions, different indexes were used, such as Shannon, Simpson, and InvSimpson, which highlight evenness, as opposed to Observed, Chao1, ACE or Fisher indexes, which are more related to richness. Different distance methods were used to estimate the community composition differences (β-diversity) between two conditions, as Bray Curtis, Jaccard and weighted Unifrac. The farm and the lactic starter dose modulation effects on alpha-diversity and beta-diversity were tested by PERMANOVA and Adonis’ tests, respectively. The differences of dispersion of bacterial profiles to the centroid between conditions were tested by TukeyHSD test. Differential analyses of families, genus and species abundances in samples, according to the farm or the dose, were performed on taxa with a minimum relative abundance of 0.001 and a minimum prevalence of 4 in the milk and of 3 in other samples. Three methods were used: DeSeq, MetaGenomeSeq and MetaCoder. A taxon was considered as differentially abundant between tested conditions when it was differentially found with a minimum of two methods (*p*-value < 0.05).

## 3. Results

### 3.1. Quality of Raw Milks

The major biochemical characteristics of the raw milk, originating from each of the two farms and used to make the cheese, are summarized in the Appendix A. None of the measured descriptors (pH, levels of lactose, fat, protein, urea, casein and the number of somatic cells) showed a significant difference according to the farm they originated from.

The level of total microorganisms tends to be higher in the milk from the farm F_15 (1.30 × 10^4^ CFU/mL) than in the milk from the farm F_38 (6.33 × 10^3^ CFU/mL) (Appendix A). The mean levels of autochthonous coagulase positive *Staphylococcus* were 1.04 × 10^2^ and 6.16 × 10^2^ CFU/mL in the raw milk from the farms F_15 and F_38, respectively, with no significant difference between the farms. The levels of total *E. coli* were lower than 10 CFU/mL whichever the farm, and no STECs were detected considering the detection threshold in the enrichment broth (<5 CFU/mL).

### 3.2. Levels of Pathogens

After the inoculation of the pathogens into the milks, mean levels of *S. aureus* were 9.30 × 10^2^ and 6.37 × 10^2^ CFU/mL in the milk from the F_15 and the F_38, respectively. The levels of *S. aureus* in most of the control cheeses were either below the detection threshold, or below those in the corresponding inoculated cheeses whatever the ripening stage. The level of *S. aureus* in the inoculated cheeses increased over the first 24 h of cheese-making (Figure 1), to reach mean levels of 1.95 × 10^6^, 3.09 × 10^5^ and 1.25 × 10^5^ CFU/g for the lactic starter doses 0.1×, 1× and 2×, respectively. At D1, no significant difference in *S. aureus* level was found in relation to the lactic starter dose, whether the farms were treated together or separately (Appendix A). In addition, none of the staphylococcal enterotoxin A to E was detected whatever the lactic starter dose (data not shown). At the end of ripening (D28), the levels of *S. aureus* in the inoculated cheeses were significantly lower than at D1, reaching 1.53 × 10^5^, 2.28 × 10^4^, and 2.33 × 10^4^ CFU/g for the lactic starter doses 0.1×, 1× and 2×, respectively (Appendix A). The level of *S. aureus* was significantly higher with the 0.1× dose than with both of the other tested doses (Figure 1), with mean differences of 0.8 Log [CFU/g] compared with both the 1× (*p*-value = 0.004) and the 2× doses (*p*-value = 0.024). No significant difference was reported either with respect to the farm of origin, the ripening time or the dose interaction. Whatever the ripening time, the levels of *S. aureus* did not differ between the 1× and 2× lactic starter doses.

After the inoculation of the pathogens into the milk, mean levels of STEC O26:H11 were 1.04 × 10^2^ and 1.13 × 10^2^ CFU/mL in the milk from the F_15 and the F_38, respectively. In the control cheeses, the STEC O26:H11 level was below the detection threshold (Appendix A), whatever the lactic starter dose, the farm or the ripening time. In the inoculated cheeses, over the first 24 h of the cheese-making process, the level of STEC O26:H11 increased to reach mean levels of 2.75 × 10^7^, 7.75 × 10^5^ and 1.96 × 10^5^ CFU/g for the lactic starter doses, 0.1×, 1× and 2×, respectively (Appendix A). At D1, the STEC O26:H11 level was higher with the 0.1× dose than with both of the other doses (Figure 2), with mean differences of 1.8 Log [CFU/g] compared with the 1× dose (*p*-value = 0.060) and of 2.3 Log [CFU/g] compared with the 2× dose (*p*-value = 0.001).

The STEC O26:H11 levels in the cheeses at the end of ripening (D28) remained stable compared with those in the cheeses at D1, with mean values of 3.44 × 10^7^, 4.20 × 10^5^ and 1.10 × 10^5^ CFU/g for the 0.1×, 1× and 2× doses, respectively (Appendix A). On D28, the STEC O26:H11 level was higher with the 0.1×dose than with both of the others, with mean difference of 2 Log [CFU/g] compared with the 1× dose (*p*-value = 0.069) and of 2.5 Log [CFU/g] compared to the 2× dose (*p*-value = 0.001). Whatever the ripening time, no difference in the levels of STEC O26:H11 was found either between the 1× and 2× lactic starter doses, or between farms.

### 3.3. Physico-Chemical Parameters

The pH values in the pathogen-inoculated cheeses and in the control cheeses were compared and no significant difference was reported, whatever the ripening time or the lactic starter dose (Appendix A). The results detailed below (Figure 3) correspond to the inoculated cheeses. Whatever the lactic starter dose, a significant decrease in the pH values was observed in the milk (with a mean value of 6.66) at D0.25, where mean values reached 6.53, 5.93 and 5.74, for 0.1×, 1× and 2× doses, respectively. Between D0.25 and D1, pH values dropped to 5.47, 5.23 and 5.19 for 0.1×, 1× and 2× doses, respectively. The pH value in the cheeses with the 0.1× dose dropped further to 5.24 at D28. The acidification was delayed with the 0.1× dose compared with both of the other doses at the early stages of the cheese-making process (D0.25 and D1). At D0.25, the pH value in the cheeses with the 0.1× dose exceeded those in the cheeses at 1× and 2× dose by 0.60 (*p*-value = 0.059) and 0.79 (*p*-value = 0.001), respectively. At D1, the pH value in the cheeses with the 0.1× dose exceeded those in the 1× and the 2× dose cheeses by 0.24 (*p*-value = 0.096) and by 0.28 (*p*-value = 0.006), respectively. At D28, the pH values did not significantly differ with respect to the lactic starter dose. However, the pH values in the cheeses manufactured with the milk from the F_15 were significantly lower than those in the cheeses manufactured with the milk from the F_38, with respective mean values of 5.21 and 5.37 (Appendix A). Whatever the ripening time, no difference in the pH values was observed between the 1× and 2× lactic starter doses.

The dry matter content in the pathogen-inoculated cheeses and in the control cheeses were compared and no significant difference was reported, whatever the ripening time or the lactic starter dose (Appendix A). The results detailed below correspond to the inoculated cheeses. At D1, a lower dry matter content was observed with the 0.1× dose compared to the 2× dose, and then it remained stable between D1 and D28 in the 0.1×-dose cheeses, but significantly increased in the cheeses manufactured with 1× and 2× doses. No difference in dry matter content was found between either the 1× and 2× lactic starter doses, or in relation to the farm, whatever the ripening time.

The Young’s modulus values (Appendix A), obtained from rheological analyses, were significantly lower with the 0.1× dose than with the 2× dose of the lactic starter, which involved the higher the lactic starter dose, the firmer the cheese. Difference relating to the farm was also observed, and the mean Young’s modulus value in cheeses manufactured with milk from the F_38 was significantly higher (mean value of 0.401 MPa) than in cheese manufactured with milk from the F_15 (mean value of 0.327 MPa).

### 3.4. Microbial Counts

At the end of ripening time (D28), the levels of thermophilic streptococci in the cores of the control cheeses (Appendix A) were significantly higher in the cheeses manufactured with the 2× dose of the lactic starter than those with the 0.1× dose (*p*-value = 0.044). In the cheese cores, no significant difference was observed with respect to the farm. In the rinds, no significant difference in the level of thermophilic streptococci was observed, with respect to the lactic starter dose, and mean levels reached 2.66 × 10^8^ CFU/g. However, higher levels of thermophilic streptococci were noticed in the cheeses manufactured with the milk from the F_38 than in those manufactured with the milk from the F_15.

The levels of the other microbial groups (Appendix A) were similar whatever the lactic starter dose, except that the higher the lactic starter dose the lower the ripening bacteria level in the cheese cores. Nevertheless, several microbial groups showed different levels in the cheeses in relation to the farm: the levels of the yeasts and of ripening bacteria varied both in the cheese cores and in the cheese rinds, while the levels of molds and of lactic bacteria differed only in the cheese cores or in the cheese rinds, respectively.

### 3.5. Bacterial Diversity

The differences in the bacterial community evenness and diversity, according to the farm of origin and to the lactic starter dose, were evaluated. The bacterial community in the milk and in the cheese cores differed in richness depending on the farm they originated from, as indicated by the Observed, Chao1, ACE and Fisher indexes (Appendix A). Milk from F_15 showed a higher richness than that from the F_38 but, during the ripening, for all lactic starter doses combined, the difference was reversed, with the richness index higher with the F_38 than with the F_15. Considering the data of the two farms combined, the most significant differences in the bacterial community richness and evenness were linked to the lactic starter dose, especially in the cheese cores (*p*-value ≤ 0.001). The values of the richness and the evenness indexes were increased when the lactic starter dose was decreased, whatever the index (Shannon, Simpson or InvSimpson) mainly in the cheese cores (D8 and D28) and, to a lesser extent, in the cheese rinds. These indexes, whatever the dose or the farm, also increased between D8 and D28 in the cheese cores.

The ordination of the bacterial community profiles through non-metric multidimensional scaling (NMDS) according to the farm or to the lactic starter dose was evaluated. The bacterial profiles in the raw milk (Appendix A), and in the cheese cores at D28 (Figure 4D, Appendix A), were clustered by farm, as shown by the Bray-Curtis, Jaccard and wUF methods. The variability in the bacterial composition of the milk (Appendix A) was higher in the milk originating from the farm F_38 than in that from the farm F_15, as shown by the difference in profiles dispersion to centroid. The bacterial profiles of the cheeses were clustered principally by lactic starter dose (Figure 4), as shown by the Bray-Curtis, Jaccard and wUF methods. We noted a cluster composed of the cheeses manufactured with the 0.1× dose, significantly distant from both of the other doses, in the cheese cores (at D8 and D28) and, to a lesser extent, in the cheese rinds. The bacterial profiles in the cores of the cheeses (at D8 and D28), manufactured with the 0.1× dose, were also more scattered than those manufactured with both of the other doses, but the dispersion to centroid for 1× and 2× doses increased during the cheese ripening and was higher in the cheese rinds than in the cheese cores. No difference was observed between 1× and 2× dose.

The *Streptococcaceae* family, *Streptococcus* genus and *S. thermophilus* species were the most predominant bacterial taxa in the cheese cores, both at D8 and D28, as well as in the cheese rinds. However, their relative abundance decreased during ripening and was lower in the rinds than in the cores. None of the DNA sequences detected in the raw milk or in the control cheeses was affiliated to *S. aureus*, or to *E. coli*. Differential analyses were performed on the relative abundance of bacterial taxa depending on the tested conditions in the raw milk, the cheese cores at D8 and D28 and the cheese rinds. For each ripening time, the family, genus and species ranks were investigated. Mean relative abundance of taxa are displayed in the Appendix A. Only taxa differentially abundant between conditions are reported in the table. Whatever the taxonomic rank, most of the differences in relation to the farm were observed in the milk, and none was observed in the cheese rinds. In the raw milk samples, differentially abundant taxa were generally more abundant in the F_15 than in the F_38. Thus, the *Streptococcaceae* family, in this case mainly represented by the species *Lactococcus lactis*, were more abundant in the raw milk from the F_15, followed by the *Lactobacillaceae*, *Bifidobacteriaceae* and *Corynebacteriaceae* families. Conversely, the higher mean relative abundance of species and genus belonging to the *Pseudomonadaceae* and *Bacillaceae* families and the *Streptococcus* genus were observed in the raw milk samples from the F_38. At D8, only the *Lactobacillaceae* family, genus and species still showed higher mean relative abundance in the F_15 cheeses. In addition, the *Staphylococcus* genus (identified as *S. xylosus*, a non-CoPS species) and *Enterococcus faecalis* presented higher mean relative abundance in the F_38. Finally, at D28, additional taxa belonging to *Enterococcaceae*, *Enterococcus*, *E. faecalis* and *Serratia* sp. showed higher mean relative abundance in the cores of the F_38 cheeses.

Differences in the relative abundance of some families, genera and species were observed with respect to the lactic starter dose in the cheese cores, whatever the time of ripening, as well as in the cheese rinds. The relative abundance of the *Streptococcaceae* family, *Streptococcus* genus and *S. thermophilus* increased with the lactic starter dose in the cheese cores both at D8 and D28, with a maximum difference of 1.8-fold between the 0.1× and the 2× doses at D28. The same trend was observed in the cheese rinds, but with a higher difference in mean relative abundance between the 0.1× dose and both of the other doses (with multiplying factors of 10× and 16× for the 1× and 2× doses, respectively). The families, genera and species belonging to the *Enterobacteriaceae* and *Enterococcaceae* families and to *L. lactis* were differentially abundant in relation to the lactic starter dose at D8. Their mean relative abundance was significantly higher with the 0.1× dose than with both of the other doses. Then, at D28 in the cheese cores, the mean relative abundance of *Enterobacteriaceae* and *Enterococcaceae* families dropped to an equivalent abundance regardless of the lactic starter dose. On the contrary, the mean relative abundance of *L. lactis* increased between D8 and D28 and was still significantly higher with the 0.1× dose than with both of the other doses. In the cheese rinds, the higher the dose, the lower the mean relative abundance of *Brevibacterium* genus and species, of the *Enterobacteriaceae* and, to a lesser extent of the *Staphylococcaceae* families, with significant differences between the 0.1× and the 2× doses.

## 4. Discussion

Managing sanitary protection while promoting microbial diversity is a major challenge for raw milk cheese manufacture improvement. Several studies have reported differences in the microbial or physico-chemical characteristics of raw milk cheeses with or without added lactic starter. To our knowledge, few studies have evaluated the impacts on acidification of modulating the lactic starter dose from the recommended one, or on the level of pathogens or the microbial diversity in the cheese. We hypothesized that adjusting the dose of lactic starter used could be a key to managing sanitary protection while promoting bacterial diversity in raw milk cheeses. We compared the impacts of the lactic starter used as recommended by the manufacturer with those of a 10-fold lower and a 2-fold higher dose on the levels of *S. aureus* and STEC O26:H11 and on the microbial diversity in uncooked pressed cheese made from raw milk. We tested a high-risk situation, combining high levels of inoculated pathogens and a cheese process which could allow their growth. The levels of pathogens were defined according to previous studies in literature, allowing for comparisons between growth patterns: initial inoculation levels of 2 Log [CFU/mL] of STEC O26:H11 were used to describe its growth in uncooked pressed model cheeses [47,48]. Medved’ová et al. [49] studied the impact of lactic starter addition on growth of *S. aureus* and *E. coli* at 3 Log [CFU/mL] each, in pasta-filata cheeses.

We observed a delay in acidification with the 0.1× dose compared to both the 1× and the 2× doses, with differences of 0.60 after 6 h (D0.25) and of 0.24 after one day (D1). The 10-fold reduction in the lactic starter dose was associated with a less significant delay in acidification compared to that observed without a lactic starter added, as reported in the literature. Thus, Carafa et al. [24] reported pH values reduced by 0.39 to 0.53 at 24 h in Malga Traditional Mountain cheeses (an Italian semi-hard cheese manufactured with raw milk) prepared with an autochthonous thermophilic starter added at 5 × 10^6^ CFU/mL, compared with cheeses without the addition of the lactic starter. An even larger reduction in the pH was observed after 24 h (0.6 units) by Medved’ová et al. [49], between the cheeses manufactured with a thermophilic starter added at 10^5^ CFU/mL and the control pasta-filata cheeses without a lactic starter added.

In our study, no significant difference in pH values was observed between the cheeses manufactured with the 1× or the 2× dose. These results are consistent with those reported in the work of Khosrowshahi et al. [30], which studied the impact of three different doses (1× as the reference, 2× and 4×) of a mesophilic lactic starter on acidification during the manufacture of Iranian White Cheese. They observed reductions in pH values at renneting by 0.08 and 0.28 between the 1× dose and the 2× or 4× doses, respectively.

Moreover, the recommended dose and the 2× dose allowed Saint-Nectaire-like cheeses to reach targeted pH value at D0.25 (around 5.8, or below, at 6 h), which limits the growth of CoPS at 24 h [15], conversely to the 0.1× dose. This result highlighted the importance of the recommended dose as a minimal lactic starter dose to ensure correct acidification, possibly contributing to the management of sanitary risks.

To our knowledge, as regards both pathogens evaluated in the present study, the scientific literature has only reported their behavior in cheeses manufactured with or without the addition of a lactic starter.

Although the initial level of STEC can influence the maximal level reached in cheese, a similar growth pattern was reported by different authors. In our study, we observed a fast increase in the STEC level by around 4 Log [CFU/g] during the first 24 h. This is similar to the results reported by Callon et al. [48], Delbès-Paus et al. [47], or Frétin et al. [34], in uncooked pressed cheese models with a standard dose of thermophilic starter (around 10^7^ CFU/mL) and initial STEC inoculation levels of either 0.5 or 2 Log [CFU/mL]. We observed a persistence until 28 days of ripening, whatever the tested lactic starter dose. This was also observed by the abovementioned authors. Yet some studies on different cheese types, with longer duration of ripening from 15 weeks to 7 months, showed a reduction in the level or even a non-detection of *E. coli* at the end of ripening [24,28]. At D1 in our study, decreased levels of STEC O26:H11 by 1.8 Log [CFU/g] and 2.3 Log [CFU/g] were observed with 1× and 2× doses compared with the 0.1× dose, respectively. The same trend was observed at D28, with levels of STEC O26:H11 decreased by 2 Log [CFU/g] and 2.5 Log [CFU/g], with the 1× and the 2× doses, respectively, compared with the 0.1× dose. The differences in STEC counts in cheese at D1 fall into the same range as those reported by Medved’ová et al. [49], showing a decrease in the levels of *E. coli* by 2.10 Log [CFU/g] at D1 in cheeses with the addition of a lactic starter in comparison with the control cheeses with no lactic starter added.

The growth patterns of *S. aureus* in our uncooked pressed cheese models were consistent with those previously reported. Delbès et al. [15] described a fast growth of *S. aureus* during the first 6 h of the cheese-making process (+3 Log [CFU/g]) and then a slight reduction during ripening (−0.55 Log [CFU/g] at D30). In addition, we showed that the *S. aureus* levels were increased by 0.8 Log [CFU/g] at D28 in the cheeses manufactured with the 0.1× dose compared with those manufactured with both of the other doses. The threshold of 10^4^ CFU/g for CoPS counts in cheeses products, as defined by the Commission Regulation (EC) No 2073/2005 of 15 November 2005 on microbiological criteria for foodstuffs (last update: 8 March 2020) [50], was consequently overpassed with the 0.1× lactic starter dose, which was not the case with the recommended dose. *S. aureus* SA15 produced staphylococcal enterotoxins in vitro in Brain Heart Infusion incubated at 30 °C for 48 h [51] and was found to possess the enterotoxin-encoding *sec4* and *sel2* genes [52]. Yet, no staphylococcal enterotoxin was detected in the inoculated cheeses on D1, whatever the lactic starter dose. These results suggest that neither the SA15 strain, nor the *S. aureus* populations naturally present in the raw milk used, produced enterotoxins in detectable concentrations in the tested cheeses, despite *S. aureus* levels reached 6 Log [CFU/g] and pH was around 6.5 at D0.25 at the 0.1× dose. These results were consistent with those from previous studies in Saint-Nectaire-model cheese made from cows’ raw milk [15], similar to our cheese models, in which no staphylococcal enterotoxin was detected, although coagulase-positive staphylococci counts were around 5 Log CFU/g at D0.25, a propitious condition to SE accumulation. The increase in the *S. aureus* counts with the 10-fold reduced lactic starter dose are similar to those reported by Al-Nabulsi et al. [53] in white brined cheeses manufactured without a lactic starter. Using standard brine (10% of NaCl) and a storage temperature of 25 °C, the authors showed reductions by 1.17 Log [CFU/g] and by 0.56 Log [CFU/g] at D1 and at D28, respectively, between the cheeses manufactured without and with a lactic starter. Medved’ová et al. [49] also observed a reduction by 0.67 Log [CFU/g] of CoPS levels at D1 in pasta-filata cheeses with an added lactic starter compared with those without a lactic starter. At the end of storage (D28), they obtained a reduction of 2.04 Log [CFU/g] between the cheeses with and without a lactic starter.

Within the range of the lactic starter doses tested in the present work, the best protection against pathogens (as STEC O26:H11 or *S. aureus*) was obtained with the lactic starter added as recommended by the manufacturer. While we observed no difference, whatever the pathogen, between the 1× and 2× doses, a 10-fold lower dose did not guarantee protection against sanitary risks. This protecting effect of the lactic starter was associated with a faster acidification and lower pH values at D1, but it cannot be excluded that it could be related to antimicrobial properties of *S. thermophilus*. As shown by several recent studies, *S. thermophilus* strains possess anti-pathogenic properties associated with the production of bacteriocins [54]. However, we used a commercial and complex lactic starter aimed at controlling acidification in a wide range of cheese processes and subject to the manufacturer’s culture strain rotation strategy to combat phages in the dairy industry. To the best of our knowledge, no information is available on the ability of the strains included in this lactic starter to produce bacteriocins, but we think it unlikely that strains with specific anti-pathogenic properties (including production of bacteriocins) will be added in rotation in this type of lactic starter.

Despite the decreasing relative abundance of *S. thermophilus* sequences during ripening, this species was still found in high relative abundance at the end of ripening, both in the cheese cores (from 46% of the total 16S rRNA gene sequences with the 0.1× dose to 85% with the 2× dose) and in the cheese rinds (up to 75% with the 2× dose). We noticed a 10-fold lower relative abundance of *S. thermophilus* in the cheese rinds with the 0.1× dose compared with both of the other doses. It cannot be excluded that the abundance of *S. thermophilus* may be overestimated after metabarcoding analyses due to the possible amplification of DNA from dead cells. However, the counts in viable and cultivable streptococci in the cheese cores at D28 were still high, increasing with the lactic starter dose and ranging from 5.7 × 10^8^ to 1.41 × 10^9^ CFU/g. In the cheese rinds, the levels of thermophilic streptococci were not different in relation to the lactic starter dose and were lower than those obtained in the cheese core (reached 2.38 × 10^8^ CFU/g). On the whole, these results are consistent with those reported by Silvetti et al. [55] in Silter cheese (Italian hard cheese), who noticed an increase from 7.3 to 77% of *Streptococcus* spp. between curd and end of ripening (200 days) in cheeses with a thermophilic starter added at 10^6^ CFU/mL. Conversely, several authors showed a reduction or an absence of detection of the added streptococci in fresh cheese [28] and in semi-hard cheeses matured for up to 90 days [31], or even 120 days [25].

In our work, at D8, we observed that, in the cheese cores, minority taxa such as *Lacticaseibacillus*, *Staphylococcus* genera and species, or, at D28, *Enterococcus* and *Serratia* genera and species, were differentially abundant depending on the farm. However, the bacterial community composition of the cheese was mainly driven by the lactic starter dose. Reduced richness and evenness (α-diversity) or dispersion of bacterial profiles to the centroid (beta-diversity), were observed with the 1× and the 2× lactic starter dose compared with the 0.1× dose. These results were consistent with those obtained by Choi et al. [28] who described, in semi-hard Cheddar cheese, higher α-diversity index (Shannon and Chao1) without an added lactic starter than with the addition of a thermophilic starter (inoculated at 1 × 10^6^ CFU/mL). Minority taxa (<7%) affiliated to the *Enterobacteriaceae* family, which includes potential opportunistic species, showed higher relative abundance in the cheeses with the lowest amount of lactic starter. Carafa et al. [24] also observed a higher relative abundance of *Enterobacteriaceae* in their control cheeses than in the cheeses with an added lactic starter. Furthermore, Silvetti et al. [55] reported lower specific richness in cheese with the addition of an autochthonous lactic starter, including unwanted bacteria as *Pseudomonas* or *Serratia* spp. Unlike other studies, we found that the dominant taxa, corresponding to genus or species with potential interest in cheese, also showed higher relative abundance with the 0.1× dose than with both of the other doses. Notably, the relative abundance of autochthonous *L. lactis* dropped from 17% to 0.2% in the cheese cores at D8, and from 21% to 0.8% at D28, by increasing the thermophilic starter dose. Similarly, in the cheese rinds, the relative abundance of *Brevibacterium* genus and species decreased from around 26% to 4% between the 0.1× dose and the 2× dose. In the early stages, *L. lactis* contributes to acidification and flavor development through the involvement of the lactose and citrate metabolisms [3,56] and can also produce antimicrobial compounds such as nisin [57]. Some species belonging to *Brevibacterium* genus, as *B. linens*, *B. antiquum* or *B. aurantiacum*, are involved in the cheese-ripening process and participate in the development of sensorial typicity (flavor and color) [58,59], as well as in proteolysis [60]. While the counts in cultivable LAB were comparable regardless of the lactic starter dose, the counts in ripening bacteria in the cheese cores decreased as the added lactic starter dose increased (Appendix A). Given the major role of LAB, and of ripening bacteria as *Brevibacterium* sp. in cheese flavor development, the changes in their relative abundance bring into question the potential impact of the lactic starter dose on the cheese sensory qualities. Considering the slower acidification and draining of the 0.1× dose fresh cheeses, that resulted in lower dry matter content at D1 and D28, it is very likely that the lactic acid starter dose affected the ripened cheese texture and flavor. This assumption is supported by the increased Young’s modulus values, indicating an increased cheese firmness with increased lactic starter dose, and likely negatively correlated to the proteolysis level [61].

## 5. Conclusions

To sum up, the 10-fold lower lactic starter dose enhanced the bacterial diversity in the cheese as well as the relative abundance of potentially helpful taxa (*Brevibacterium* or *Lactococcus* species) along with that of other potentially opportunistic ones (*Enterobacteriaceae* family). Microbial diversity can be associated with a protecting effect against pathogens in cheese [34,62,63]. In the present study, despite an increased bacterial diversity, the delayed acidification observed with the 0.1× dose was likely responsible for the higher levels of the inoculated STEC and *S. aureus* in the cheeses.

Given that sanitary criteria were paramount to our study, it can be said that the 10-fold lower dose is to be proscribed. Nevertheless, with lower and more realistic levels of inoculated pathogens, an intermediate reduced dose of lactic starter, between the 0.1× and the standard dose, may be sufficient to establish an efficient balance between acidification, pathogen growth and microbial diversity.

No difference in the level of pathogens or in bacterial diversity in the cheese was observed between the 1× and 2× doses. Further studies would be needed to determine if this was a result of an insufficient difference between the two tested doses or from a plateau effect.

Despite the reduced dispersion of cheese bacterial patterns with an increase in the dose of the lactic starter cultures, our study revealed differences in the bacterial community composition in relation to the farm of origin, not only in the raw milk, but also in the cheeses. Since the microbiota of raw milk is highly variable from one farm to another, the effects of adjusting the lactic starter dose should be evaluated with a larger range of raw milk.

To conclude, we confirmed that the modulation of the amount of the lactic starter could be a key to managing the qualities and the sanitary risks in cheese, although specific studies would be necessary to finely adjust the dose depending on the type of the lactic starter and on the type of the cheese. Insofar as the balance of the indigenous microbial populations of potential interest for cheese production has been altered in relation to the lactic starter dose, it would be important to assess the associated effects on the sensory qualities of the cheese.

## Figures and Tables

**Figure 1 microorganisms-09-01081-f001:**
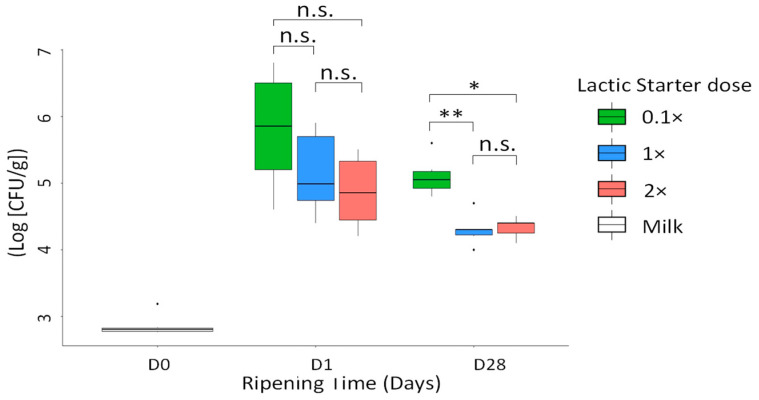
Level in Log [CFU/g] of *S. aureus* in milk (D0) and in cheeses at D1 and D28, according to the lactic starter dose (*n* = 6). The differences according to the dose were estimated by a multiple comparison test, a Dunn’s test. * *p* < 0.05; ** *p* < 0.01; n.s. non-significant. For milk samples (*n* = 6), no statistical tests were performed.

**Figure 2 microorganisms-09-01081-f002:**
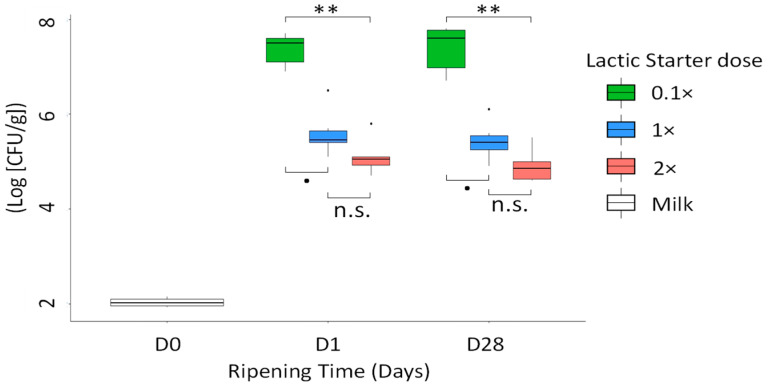
Level in Log [CFU/g] of STEC O26:H11 in milk (D0) and in cheeses at D1 and D28, according to the lactic starter dose (*n* = 6). The differences according to the dose were estimated by a multiple comparison test, a Dunn’s test. ** *p* < 0.01; ^●^ tendency; n.s. non-significant. For milk samples (*n* = 6), no statistical tests were performed.

**Figure 3 microorganisms-09-01081-f003:**
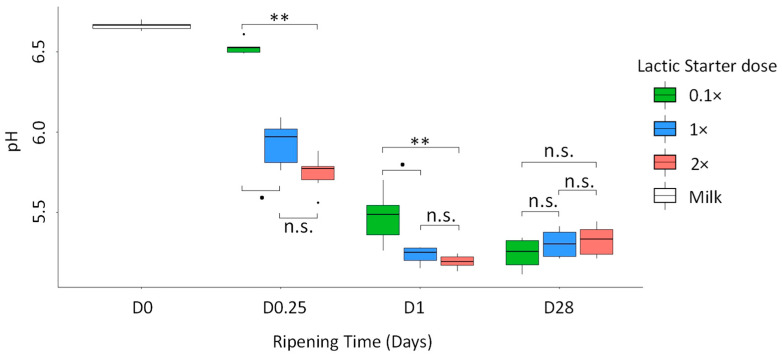
pH values in milk (D0) and in cheeses at D0.25, D1 and D28, according to the lactic starter dose (*n* = 6). The differences according to the dose were estimated by a multiple comparison test, a Dunn’s test. ** *p* < 0.01; ^●^ tendency; n.s. non-significant. For milk samples (*n* = 6), no statistical tests were performed.

**Figure 4 microorganisms-09-01081-f004:**
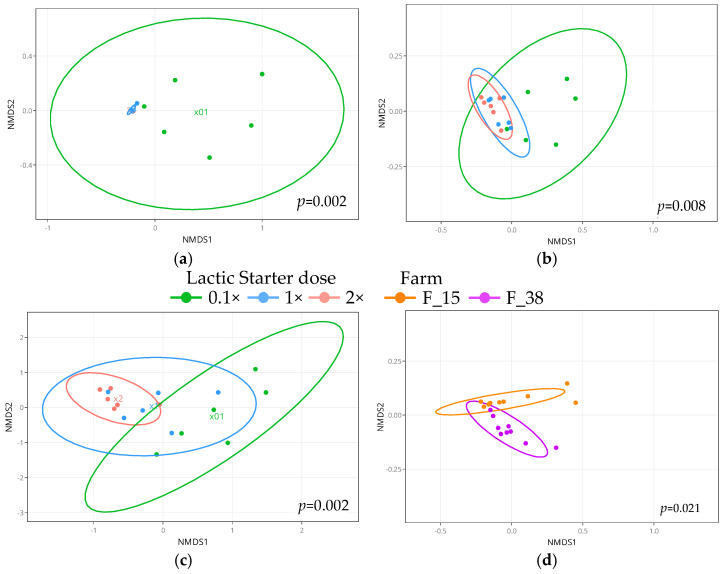
Beta-diversity of (**a**) Cheese core samples at D8 (*n* = 18), (**b**) Cheese core samples at D28 (*n* = 18), and (**c**) Cheese rind samples at D28 (*n* = 18) clustered according to the lactic starter dose; and (**d**) Cheese core samples at D28 (*n* = 18) clustered according to the farm. Non-Metric multidimensional scaling (NMDS) based on the Bray Curtis algorithm of bacterial communities. *p*-value was obtained after permutational MANOVA analysis (Adonis statistical test) and indicates significance between sample groups.

## Data Availability

Raw sequence data are available at the European Nucleotide Archive (ENA) database under the accession number PRJEB44120.

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
