# Peer review of "Lactic Starter Dose Shapes S. aureus and STEC O26:H11 Growth, and Bacterial Community Patterns in Raw Milk Uncooked Pressed Cheeses"

_microorganisms, 2021, doi:10.3390/microorganisms9051081_

Round 1
Reviewer 1 Report
General comment:
The manuscript by Piqueras et al. reports whether a dose of lactic starter influence the protection against pathogens and bacterial diversity of uncooked pressed cheese produced from raw milk. The methods are well written and the results are clearly presented and discussed. According to the presented results, the variations in the dose of the starter culture have the impact on the safety and quality of cheese made from raw milk, what is the main scientific contribution of the manuscript. However, a few of improvements are needed in the terms of clarity of the manuscript. Therefore, the manuscript is not acceptable for publication in its present state.
Specific comments:
Some S. aureus strains among natural populations of raw milk may produce enterotoxins and there are no data about potential of added S. aureus SA15 strain to produce enterotoxins. Analysis of eventual presence of enterotoxins in the produced cheese samples will be valuable.
Furthermore, the data regarding the ability of used lactic starter strains and ripening starter strains to produce bacteriocins with antagonistic effect against S. aureus in cheese during production and ripening are also missing.
As authors mentioned in the Conclusion of the manuscript, it would be important to assess the associated effects of starter dose on sensory qualities of the produced cheese samples as a consequence of the influence on the indigenous microbial populations.
Without additional data mentioned above, it is just a preliminary communication or scientific note.
Abstract:
Conclusion regarding the impacts of the dose of a thermophilic lactic starter a x2 higher than the recommended one, on acidification, growth of Staphylococcus aureus and Shiga-toxin-producing Escherichia coli (STEC) O26:H11, as well as on the bacterial community patterns is missing. Namely, the one aim of the study was “whether the protection against pathogens could be improved using a higher starter dose, and at what cost for the bacterial diversity.”
Line 18 - it would be better signed the strains used in the study Staphylococcus aureus and Shiga-toxin-producing Escherichia coli (STEC) O26:H11 with their strain marks (F43368 and SA15, respectively). Furthermore, names of microorganisms must be written italic (check the whole text of manuscript)!
Introduction:
Line 108-109 – “S. aureus and STEC 108 O26:H11” In the description of the aim of study it would be better to introduce strain marks (F43368 and SA15, respectively). Further in the text of manuscript it is not necessary.
Materials and Methods:
Line 139-140 - “The strains SA15 and F43368 were inoculated at the expected levels of 103 CFU/mL and 102 CFU/mL, respectively” Why these levels of inoculation were chosen? Furthermore, there are used only strains marks without their names, but in the rest of the manuscript the names of strains were used without strains marks. The version in the rest of the manuscript could be better.
Line 187-188. “The levels of E. coli and total germs were evaluated by Agro-187 Lab’s” – the methods must be mentioned and cited (including selective media used).
Results, Discussion, Conclusion and Figures:
It would be better to replace the term “starter” with “lactic starter” because two commercial starter cultures were used, lactic and ripening starters.
Reviewer 2 Report
The authors present an interesting report that is more of a short research note than a full research paper. For the readers that work on cheese starters, this is certainly an interesting paper. It would benefit from adding additional data, such as the impact of starter levels on staphyloccocal "enterotoxin" accumulation, and/or on the sensorial characteristics of (pathogen-free) cheeses. Also, data on the effect of the level of starter on proteolysis extent could have been added. In my opinion, the manuscript would have benefitted from a joint Results & Discussion section, rather than separating both. Otherwise, it reads well and touches (a tad too lightly) an aspect of cheesemaking that has been little studied until now. A few other minor remarks are given in the annotated version of the manuscript annexed to this review.
In my opinion, this manuscript deserves being accepted as a short research note, in which case is should be shortened. Otherwise, the authors could add the additional data suggested, so that it would become a full research paper.

Reviewer 3 Report
The aim of the research falls within the thematic scope of the journal.
The purpose of the original article was to investigate: 1) whether a reduced dose of lactic starter, compared with the recommended dose, would satisfy sanitary criteria while minimizing the effects on the microbial diversity of raw milk uncooked pressed cheese; 2) whether the protection against pathogens (S. aureus and STEC O26:H11) could be improved using a higher starter dose and at what cost for the bacterial diversity. Taking the dose recommended by the lactic starter manufacturer as a reference, Authors simultaneously tested a 10-fold lower (x0.1) and a 2-fold higher (x2) dose.
The research was planned very broadly, taking into account many variables of the examined factors. This does not raise any objections.
However, the prepared manuscript requires some changes and supplements when it comes to the description of the methodology of obtaining some results (note in the text line 180 - it concerns the results presented in Table S3) and their description in chapter "3. Results". It seems to me that the way of reporting the results of the microbiological analysis should be changed throughout the manuscript (eg. change from 1.95.106 CFU/g to 1.95 x 106 CFU/g). Also standardize the way of recording these results between the text and Tables S2, S3.
All remarks (minor ones; and including those not listed here) are marked in the text of manuscript in the review mode.
After introducing minor changes, in my opinion, the Editors may direct the manuscript for further processing.

Round 2
Reviewer 1 Report
The authors accepted all the requested remarks and answered all the issues, what contributed to the overall manuscript improvement. Therefore, the manuscript is appropriate for the publication.